# Quercetin-Induced Glutathione Depletion Sensitizes Colorectal Cancer Cells to Oxaliplatin

**DOI:** 10.3390/foods12081733

**Published:** 2023-04-21

**Authors:** Jinkyung Lee, Chan Ho Jang, Yoonsu Kim, Jisun Oh, Jong-Sang Kim

**Affiliations:** 1School of Food Science and Biotechnology, Kyungpook National University, Daegu 41566, Republic of Korea; 2Institute of Agricultural Science and Technology, Kyungpook National University, Daegu 41566, Republic of Korea; 3Department of Integrative Biology, Kyungpook National University, Daegu 41566, Republic of Korea; 4New Drug Development Center, Daegu-Gyeongbuk Medical Innovation Foundation, Daegu 41061, Republic of Korea

**Keywords:** quercetin, sulforaphane, glutathione reductase, oxidative stress, anti-cancer, oxaliplatin, apoptosis

## Abstract

Quercetin is an antioxidant phytochemical which belongs to the natural flavonoids group. Recently, the compound has been reported to inhibit glutathione reductase responsible for replenishing reduced forms of glutathione and thus leads to glutathione depletion, triggering cell death. In this study, we examined if quercetin sensitizes tumors to oxaliplatin by inhibiting glutathione reductase activity in human colorectal cancer cells, and thereby facilitates apoptotic cell death. A combined treatment with quercetin and oxaliplatin was found to synergistically inhibit glutathione reductase activity, lower intracellular glutathione level, increase reactive oxygen species production, and reduce cell viability, compared to treatment with oxaliplatin alone in human colorectal HCT116 cancer cells. Furthermore, the incorporation of sulforaphane, recognized for its ability to scavenge glutathione, in combination with quercetin and oxaliplatin, substantially suppressed tumor growth in an HCT116 xenograft mouse model. These findings suggest that the depletion of intracellular glutathione by quercetin and sulforaphane could strengthen the anti-cancer efficacy of oxaliplatin.

## 1. Introduction

Colorectal cancer ranks among the most frequently diagnosed types of cancer in developed nations [1]. In the year 2021, it was responsible for the third highest mortality in men and the second highest in women. Colorectal cancer accounts for 10.9% of all cancer deaths and kills more than 17 people per 100,000, and the incidence is continually increasing in the Republic of Korea [2].

Cisplatin, carboplatin, and oxaliplatin (OXP) are examples of platinum-based drugs that are extensively utilized for the treatment of diverse cancer types such as testicular, ovarian, and non-small cell lung cancers [3]. OXP, being a third-generation platinum drug, exhibits no cross-resistance to cisplatin and carboplatin [4,5]. Thus, this drug is specifically used in colorectal cancer that is resistant to cisplatin or carboplatin [3,5]. OXP is known to predominantly form DNA intra-strand crosslink adducts, cause DNA deformation, and disrupt DNA replication and repair, resulting in a cytotoxic effect [6]. Therefore, the drug has become a standard in the management of colorectal cancer [7]. Unfortunately, the intrinsic or acquired resistance of tumors to OXP hampers a complete remission of tumors. Molecular mechanisms associated with OXP resistance include increased efflux and upregulation of resistant genes [8,9]. Consequently, it is important to find a way to overcome the drug resistance to improve and optimize treatments [10,11].

In recent times, therapeutic properties such as anti-oxidant, anti-inflammatory, antimicrobial, and anti-cancer activities have been attributed to flavonoids present in a range of fruits and vegetables. Among them, quercetin (QUE) stands out as it is one of the most commonly occurring flavonoids in natural diets [12]. QUE is a naturally occurring compound that possesses a flavone nucleus comprising two benzene rings linked via a heterocyclic pyrone ring [13]. A series of studies showed that QUE can act as a glutathione reductase (GR) inhibitor [14,15].

Conversely, compared to normal cells, cancer cells exhibit elevated levels of reactive oxygen species (ROS). As a result, cancer cells have developed a variety of antioxidant defenses, including increased expression of repair and detoxifying enzymes, and small scavenger molecules such as glutathione (GSH) [16]. GSH is a tripeptide consisting of glutamate, cysteine, and glycine, and is oxidized into oxidized glutathione (GSSG) in the presence of ROS and toxic chemicals. Oxidized glutathione is regenerated into a reduced form by GR, and the reduced glutathione can be utilized in detoxification of toxic compounds [17]. Unfortunately, when a tumor has been established, elevated levels of GSH could protect cancerous cells from chemotherapeutic drugs by facilitating detoxification [18,19]. Furthermore, GR has been reported to be significantly upregulated in all clinical stages of colorectal cancer patients as compared to normal people [20].

The inhibitory effect of QUE on GR is anticipated to decrease the intracellular level of GSH in tumor cells, which, in turn, could boost the anti-cancer effectiveness of chemotherapy agents. This study was carried out based on the hypothesis that QUE could enhance the chemotherapeutic impact of OXP by increasing intracellular ROS through its inhibitory activity against GR.

## 2. Materials and Methods

### 2.1. Chemicals

Quercetin (QUE; Sigma-Aldrich, Inc., St. Louis, MO, USA), oxaliplatin (OXP; LC Laboratories, Woburn, MA, USA), and sulforaphane (SFN; LKT Laboratories, Inc., St. Paul, MN, USA) were dissolved in dimethyl sulfoxide (DMSO; Sigma-Aldrich, Inc.) before use. Unless otherwise stated, all chemicals were obtained from Sigma-Aldrich, Inc.

### 2.2. Cell Lines and Culture Conditions

The human colorectal carcinoma HCT116 cell line was obtained from the Korean Cell Line Bank (Seoul, Republic of Korea). The cells were cultivated in a maintenance medium which consisted of Dulbecco’s Modified Eagle Medium (DMEM, Welgene, Gyeongsan, Republic of Korea) supplemented with 10% (*v*/*v*) fetal bovine serum (FBS, Welgene), 2% (*v*/*v*) 4-(2-hydroxyethyl)-1-piperazine ethanesulfonic acid (HEPES, Gibco Thermo Fisher Scientific Inc., Waltham, MA, USA), 1% (*v*/*v*) Minimum Essential Medium containing a 10 mM concentration of each non-essential amino acid (MEM-NEAA, Gibco Thermo Fisher Scientific Inc.), and 1% (*v*/*v*) penicillin–streptomycin (HyClone Laboratories Inc., Chicago, IL, USA). The cells were maintained in a cell culture incubator (MCO-19 AIC, Sanyo, Osaka, Japan) at 37 °C and 5% CO_2_. The culture medium was refreshed every three or four days, and the cells were subcultured using phosphate-buffered saline (PBS, Gibco Thermo Fisher Scientific Inc.) and 0.05% (*w*/*v*) trypsin-EDTA solution (Welgene), and then plated onto culture dishes (SPL Life Sciences co., Ltd., Pocheon, Republic of Korea).

### 2.3. Cell Viability Assay

To evaluate cell viability, HCT116 cells were seeded in a 96-well plate (SPL Life Sciences co., Ltd.) at a density of 5 × 10^3^ cells per well. Following treatment with QUE in the absence or presence of OXP for 3 or 6 days, the medium was removed and CCK solution (D-Plus^TM^ CCK cell viability assay kit, Dongin Biotech., Seoul, Republic of Korea) was added to each well [21,22]. The plate was incubated in the cell culture incubator for 1 h. The absorbance at 450 nm was detected using a microplate reader (Sunrise^TM^, Tecan Group Ltd., Männedorf, Switzerland).

### 2.4. GR Activity Assay

Intracellular GR activity was examined using a commercially available assay kit (Cat. # ab83461; abcam, Cambridge, UK). HCT116 cells were plated in a 90 mm diameter culture dish (SPL Life Sciences co., Ltd.) at a density of 1 × 10^6^ cells. Following 72 h of treatment with QUE (25 μM) in the absence or presence of 0.5 μM OXP and 3 μM SFN, the cells were detached and concentrated to a density of 1 × 10^6^ cells in 25 μL of PBS. A 100 μL of cold assay buffer was added to the cell suspension and vortexed for lysis. The supernatant from the cell lysate was used for the assay as instructed by the manufacturer. After the reaction was completed, the absorbance at 405 nm was detected using a microplate reader (Sunrise^TM^, Tecan Group Ltd.).

### 2.5. GSH Level Assay

To quantify the intracellular GSH level, a commercially available assay kit (Cat. # EIAGSHC; Thermo Fisher Scientific, Waltham, MA, USA) was used. HCT116 cells were plated in a 90 mm diameter culture dish (SPL Life Sciences co., Ltd.) at a density of 1 × 10^6^ cells. Following 72 h of treatment with QUE (25 μM) in the absence or presence of 0.5 μM OXP and 3 μM SFN, the cells were detached, collected, and used for the GSH quantification assay as instructed by the manufacturer. The absorbance at 405 nm was detected using a microplate reader (Sunrise^TM^, Tecan Group Ltd.).

### 2.6. Measurements of Intracellular ROS

Intracellular ROS levels were measured as previously described [23,24]. HCT116 cells were plated in a 96-well plate with a black bottom (Nunc, Rochester, NY, USA) at a density of 1 × 10^4^ cells per well. After 24 h, the cells were pre-incubated with 20 μM 2′,7′-dichlorodihydrofluorescein diacetate (H_2_DCFDA) for 30 min, and then treated with QUE, OXP, SFN, and *tert*-butyl hydroperoxide (*t*BHP) for an additional 30 min. The fluorescence at 485 nm excitation and 535 nm emission was subsequently measured using a fluorescence microplate reader (Infinite 200, Tecan Group Ltd.).

### 2.7. Animal Experimental Design

Male BALB/c nude mice (6-week old), whose average body weight (BW) was 20 ± 2 g, were obtained from Orient Bio Inc. (Seongnam, Republic of Korea). Mice were housed under the standard conditions (temperature, 20–24 °C; relative humidity, 45 ± 5%; 12:12 light-dark cycle), and provided with ad libitum access to drinking water and standard mouse chow pellets (Daehan Bio Link, Eumseong, Republic of Korea). The animal experiment was conducted in compliance with the guidelines of the Committee on Care and Use of Laboratory Animals of the Kyungpook National University (Approval Number, 2021-0165).

To generate HCT116 xenografts in BALB/c nude mice, 3 × 10^6^ HCT116 cells were subcutaneously injected into the left and right flanks of each mouse [21,22]. After palpable tumors were developed (approximately 80 mm^3^ in volume), mice were randomly allocated into five groups (4 xenografts per group; Table 1). The experimental groups consisted of the following: (1) vehicle-treated (control), (2) SFN only at a dose of 5 mg/kg BW, (3) QUE at 50 mg/kg BW in combination with SFN, (4) OXP at 10 mg/kg BW in combination with SFN, and (5) QUE and OXP in combination with SFN.

The vehicle consisted of 10% (*v*/*v*) DMSO and 5% (*v*/*v*) Tween^®^ 80 (Sigma-Aldrich Inc.) in sterilized saline. QUE, OXP, and SFN was prepared in the vehicle, and they were administered intraperitoneally (i.p.) to the mice three times a week for three weeks (Figure 1). The tumor size was monitored regularly (three times a week) throughout the entire experimental period. After sacrificing the mice, HCT116 xenografts were removed, weighed, and frozen in liquid nitrogen for subsequent analyses.

### 2.8. Measurement of Body Weight and Tumor Growth

The BW of each mouse and the volume of the xenografted tumor were recorded three times a week until the time of sacrifice. A caliper (Mitutoyo, Kawasaki, Japan) was used to measure the length and width of each tumor. The tumor volume was calculated using the following formula: Tumor volume (mm^3^) = (Length (mm) × Width (mm)^2^)/2.

### 2.9. Western Blotting

To prepare the tumor tissue lysates, the dissected tumors were homogenized in a pre-cooled lysis buffer containing protease inhibitor (Roche Diagnositics, Mannheim, Germany) using a tissue homogenizer (Omni International, Kennesaw, GA, USA). The homogenates were fractionated as previously described [21,22]. An amount of 40 μg protein was separated on sodium dodecyl sulfate (SDS) polyacrylamide gels and transferred to polyvinylidene fluoride (PVDF) membranes (Millipore, Burlington, MA, USA). The membranes were then blocked with 1% bovine serum albumin (BSA) and incubated with primary antibodies at 4 °C for 16 h, followed by corresponding secondary antibodies. The primary antibodies used in this study were as follows: rabbit anti-poly-ADP ribose polymerase (PARP, Cell Signaling Technology, Beverly, MA, USA), rabbit anti-caspase-3 (Cell Signaling Technology) and mouse anti-cytochrome C (AbFrontier, Seoul, Republic of Korea), mouse anti-Bax (Santa Cruz Biotechnology, Inc., Dallas, TX, USA), mouse anti-Bcl-2 (Santa Cruz Biotechnology), mouse anti-β-actin (Santa Cruz Biotechnology, Inc.). The secondary antibodies were anti-rabbit IgG or anti-mouse IgG conjugated to horseradish peroxidase (Santa Cruz Biotechnology, Inc.). Protein bands were visualized using SuperSignal^TM^ West Pico Plus Chemiluminescent and Femto Maximum Sensitivity Substrate (Thermo Fisher Scientific, Waltham, MA, USA). The band images were digitalized and analyzed using Image Quant LAS 4000 mini (GE Healthcare Life Sciences, Little Chalfont, UK) and Image Studio Lite version 5.2 (LI-COR Biotechnology, Lincoln, NE, USA), respectively.

### 2.10. Reverse-Transcription-Quantitative PCR (RT-qPCR)

The mRNA expression levels of genes encoding ATP-binding cassette subfamily G member 2 (ABCG2) and multidrug resistance protein 4 (MRP4) in HCT116 xenograft tumor tissues were analyzed using RT-qPCR [25]. Tumor samples (5 mg per tumor) were obtained from the sacrificed mice. Total RNA was extracted from the tumor lysates using Takara MiniBEST Universal RNA Extraction Kit (Takara Bio Inc., Kusatsu, Shiga, Japan), and quantified using a Nanodrop spectrophotometer (Thermo Fisher Scientific Inc.). The cDNA was synthesized by reverse-transcribing the total RNA using a cDNA Synthesis Kit (Thermo Fisher Scientific Inc.) with Oligo dT (Enzynomics, Daejeon, Republic of Korea). The qPCR mixture containing the synthesized cDNA, TOPreal™ qPCR 2xPreMIX (SYBR Green with low ROX) (Enzynomics), and forward and reverse primers of the targeted gene was prepared, followed by running the reaction in LightCycler^®^ 96 System (Hoffmann-La Roche AG, Basel, Switzerland). The acquired qPCR data were analyzed using LightCycler^®^ 96 software (version 1.1, Roche AG). The relative expression level of the gene of interest was normalized to the level of *GAPDH*.

### 2.11. Statistical Analysis

The collected data was analyzed using SPSS Statistics 22 software (SPSS Inc., Chicago, IL, USA). One-way analysis of variance (ANOVA) and Duncan’s multiple range test were used to determine statistical significance. Student’s unpaired *t*-test was used for comparisons between two groups, with significance indicated by *p*-values less than 0.05 or 0.01. Statistical differences were denoted by different alphabetical letters, asterisks, or hashes.

## 3. Results

### 3.1. QUE Enhanced OXP-Mediated Suppression of HCT116 Cell Proliferation

To evaluate the potential cytotoxicity of QUE or OXP, HCT116 cells were plated in a 96-well plate at a density of 5 × 10^3^ cells per well and treated with different concentrations of QUE (0, 12.5, 25, 50, and 150 μM) or OXP (0, 0.125, 0.25, 0.5, and 1.5 μM). After incubation for 3 days or 6 days, the cell viability was examined using a CCK cell viability assay (Figure 2). QUE alone was not cytotoxic at ≤25 μM for 3 days and at ≤50 μM for 6 days (Figure 2A_1_,B_1_). Meanwhile, OXP alone at ≥0.5 μM for at least 3 days affected the viability of HCT116 cells (Figure 2A_2_,B_2_; 81.3% at 0.25 μM and 55.4% at 0.5 μM for 3 days; 73.0% at 0.25 μM and 30.1% at 0.5 μM for 6 days). Interestingly, the cells exposed to QUE at 25 μM (no cytotoxic level) were found to be more susceptible to OXP treatment compared with the cells treated with OXP alone (Figure 2B_3_).

### 3.2. QUE Enhanced OXP-Induced ROS Production in HCT116 Cells

To determine if treatment with QUE can influence intracellular ROS levels, HCT116 cells were incubated with QUE (25 μM), OXP (0.5 μM), or a combination of both (25 μM QUE plus 0.5 μM OXP) (Figure 3A). Although OXP treatment significantly increased ROS production compared with the untreated control, it was less effective at generating ROS in cells than QUE. Moreover, the combination of QUE and OXP induced a more significant increase in intracellular ROS levels than OXP alone, regardless of the presence of SFN (Figure 3B).

### 3.3. QUE Inhibited GR Activity in HCT116 Cells

Based on the studies demonstrating that QUE acts as a GR inhibitor [14,15], the effect of QUE on the intracellular GR activity was examined. HCT116 cells were treated with QUE (25 μM), OXP (0.5 μM), or their combination for 72 h (Figure 4). We found that treatment with QUE or OXP significantly inhibited GR activity (Figure 4A). Notably, combined treatment with QUE and OXP further reduced GR activity. Moreover, the extent of inhibition of GR activity by QUE, OXP, or their combination was conspicuous in the presence of SFN at 3 μM which was non-cytotoxic and in fact proliferative in HCT116 cells [22] (Figure 4B). This finding indicates that QUE inhibited intracellular GR activity in HCT116 colorectal cancer cells and thus could enhance the anti-tumor activity of OXP.

### 3.4. Combined Treatment of QUE and OXP Decreased Intracellular GSH Level in HCT116 Cells

As QUE was found to inhibit GR activity, it was expected that the intracellular GSH level would be decreased or depleted in QUE-treated cells. In order to examine the effects of the compound on the GSH level, HCT116 cells were exposed to QUE (25 μM), OXP (0.5 μM), and both (25 μM QUE plus 0.5 μM OXP) (Figure 5). Combined treatment with QUE and OXP significantly reduced GSH levels in HCT116 cells, although the individual compound did not have significant effect on intracellular GSH level (Figure 5A). It indicates that a combination of QUE and OXP are more effective in lowering the intracellular GSH level than the individual treatment in HCT116 cells. This observation was clearly manifest in the presence of SFN (Figure 5B).

### 3.5. Combined Treatment of QUE and OXP Inhibited SFN-Induced HCT116 Xenograft Tumor Growth in Mice

To assess the potential of a combined treatment with QUE and OXP for tumor suppression, we established HCT116 xenograft tumors in BALB/c nude mice. The tumor growth was accelerated by SFN treatment [22]. Consistent with our previous study, the average tumor volume was considerably increased when SFN alone was administered than that in the untreated control group (Figure 6). In addition, treatment with either QUE or OXP tended to suppress SFN-induced xenograft tumor growth (Figure 6 and Figure 7).

Most importantly, the averaged tumor size, which was increased with SFN treatment, was significantly reduced when we treated the tumor-bearing mice with a combined treatment of QUE and OXP, compared with the individual treatment. These findings suggest that treatment with QUE in combination with OXP has superior efficacy in suppressing tumor growth.

### 3.6. Combined Treatment of QUE and OXP Increased Pro-Apoptotic Protein Expression

To investigate the cytotoxic effect of the combined treatment of QUE and OXP in SFN-treated xenograft tumors, we examined the expression levels of apoptosis-associated proteins in the xenograft tumors.

We found that SFN treatment had no significant effect on the expressions of apoptotic proteins in the tumors. However, the combined treatment of QUE and OXP in SFN-treated mice significantly induced apoptosis in the xenograft tumors (Figure 8). This was evident by the increased expression levels of pro-apoptotic protein Bax over anti-apoptotic protein Bcl-2 (Figure 8A), cytoplasmic cytochrome C (Figure 8B), and cleaved PARP normalized to full-length PARP (Figure 8C). However, we did not observe a significant effect of the combined treatment on the expression level of cleaved caspase 3 (Figure 8D) in the tumor xenografts. These results demonstrate that the combination of QUE and OXP effectively induces apoptosis in xenograft tumors by upregulating the expression of key apoptosis-associated proteins.

### 3.7. Combined Treatment of QUE and OXP Suppressed Drug-Resistance Genes in Xenograft Tumor

OXP, a commonly used chemotherapy drug, is metabolized through conjugation, whereby it binds with GSH in the presence of the enzyme glutathione transferase (GST). After conjugation, the drug is transported out of the cell by two efflux transporters, ABCG2 and MRP, which are known to play a role in drug resistance [26,27].

In our study, we observed that QUE in combination with OXP synergistically inhibited tumor cell growth. To investigate the mechanism behind this synergy, we examined whether QUE could inhibit the drug efflux transporters ABCG2 and MRP, or their gene expression (Figure 9). The RT-qPCR results showed that OXP induced *ABCG2* expression and QUE significantly reduced the mRNA levels of *ABCG2* (Figure 9A) and *MRP4* (Figure 9B). These findings suggest that QUE enhances the anti-cancer activity of OXP by inhibiting the expression of *ABCG2* and *MRP4*, thus suppressing the efflux of OXP. Notably, the expression of *MRP4* was not affected by SFN, while it was inhibited by QUE or QUE in combination with OXP (Figure 9).

## 4. Discussion

QUE and SFN are well-known phytochemicals and potent antioxidants that are widely recognized as a significant dietary source of flavonoids and isothiocyanates [28,29]. The compounds are rich in fruits and vegetables including onions, berries, broccoli, Brussels sprouts, and asparagus. In the current study, a combination of QUE and SFN was found to inhibit HCT116 xenograft tumor growth in synergy with OXP by inhibiting GR activity, lowering GSH level, increasing cellular oxidative stress, and thereby inducing apoptotic cell death.

GR is an enzyme that helps to prevent oxidative damage by converting the oxidized form of GSH (GSSG) to its reduced form, which is important for cellular redox balance [30,31]. QUE can reportedly act as a GR inhibitor [15,32], and we consistently found that QUE inhibited GR activity and increased the level of ROS in HCT116 cells. In combination with OXP, QUE significantly lowered the intracellular level of GSH and inhibited the proliferation of HCT116 human colorectal cancer cells, although QUE alone did not affect the GSH level or the cell proliferation. In addition, a combination of QUE and SFN was found to further enhance the inhibitory effect of OXP on the growth of HCT116 xenograft tumors in a mouse model.

OXP is a commonly used chemotherapeutic agent that inhibits the growth of colorectal cancer by interfering with the replication and transcription of DNA [7]. It belongs to the group of platinum-containing anti-cancer agents, which includes well-known drugs such as cisplatin and carboplatin. The mechanism of action of OXP makes it a valuable treatment option for patients with colorectal cancer. The pharmacokinetics of OXP involves three phases: distribution, elimination, and redistribution, with a brief initial distribution phase and a prolonged terminal elimination phase [33,34,35]. Within a cell, OXP interacts mainly with DNA, as well as RNA and proteins, resulting in the formation of interstrand adducts between adjacent guanine residues or guanine and adenine. This process interferes with DNA replication and transcription, leading to cell death [6,36].

However, resistance to OXP, either intrinsic or acquired, is a significant cause of treatment failure [37,38]. The resistance is often associated with the overexpression of certain genes, including ATP-binding cassette (ABC) transporters [39,40,41]. The ABC transporter family has seven subfamilies, A to G, based on sequence similarity [42,43]. Among these transporters, ABCB1, ABCC1, and ABCG2 are involved in the development of multidrug resistance (MDR) [44,45]. Of ABCC subfamily members, also known as multidrug resistance proteins (MRP), MRP1 to 9, are involved in the efflux of endogenous substances and xenobiotic compounds in eukaryotic cells [46,47,48]. Our results showed that OXP treatment increased the expression of *ABCG2*, but a combined treatment with QUE prevented this effect. Additionally, QUE treatment reduced *MRP4* expression, even in combination with OXP. This suggests that QUE treatment may help to reduce the risk of developing drug resistance to OXP.

Platinum-based anti-cancer drugs, including OXP, can lead to the production of intracellular ROS in both direct and indirect manners and further oxidative stress-induced cytotoxicity [49,50,51,52]. Nonetheless, the majority of cancer cells exhibit elevated levels of ROS due to an augmented metabolic rate, genetic mutation, and a state of relative hypoxia [53,54], which is attributed as a cause of metabolic reprogramming [55,56]. The cancer cell survival signaling cascade is often activated in these adapted cancer cells, which fuels cancer progression [57,58,59]. In this study, we found that QUE in combination with OXP prominently depleted intracellular GSH level in HCT116 cells and increased apoptosis in xenograft tumors. It is conceived that QUE perturbed the redox balance and thus enhanced the efficacy of OXP in cancer cells.

SFN has been reported to induce apoptotic cell death and cell cycle arrest in several tumor cell lines and to impede tumor growth, metastasis, and angiogenesis in animal models [60,61,62]. The phytochemical is also well known to induce strongly intracellular antioxidant and phase 2 detoxifying enzymes via Nrf2/ARE signaling pathway [63,64,65]. Considering that Nrf2 has a paradoxical role with both anti-oncogenic and pro-oncogenic activities [66,67,68,69], the biological effectiveness of SFN remains controversial. Our findings from an in vivo study showed that SFN promoted xenograft tumor growth in mice. SFN in combination with QUE synergistically reduced the growth in OXP-treated mice. This phenomenon is consistent with a significant increase in the production of ROS and a decrease in the expression of drug resistance genes such as *ABCG2* and *MRP4*. SFN has also been reported to deplete intracellular GSH and thus increase the chemosensitivity of cancer cells [70]. Thus, it suggests that QUE treatment in combination with SFN would be an effective strategy for sensitizing colorectal cancer cells to chemotherapeutic agents such as OXP.

However, this study has some limitations. First of all, the doses of QUE and SFN used in the study are relatively high, so that the optimum dose should be determined to be employed in human cancer therapy. Second, QUE and SFN are rapidly metabolized as soon as they are infused into the body, so a pharmacokinetic study should be conducted to validate their usefulness as a chemotherapeutic agent. Third, the precise action mechanisms of QUE and SFN in combination treatment with OXP should be further elaborated although this study elucidated a couple of possible mode of action.

## 5. Conclusions

In the present study, we have shown that QUE has an inhibitory activity against GR, can lower intracellular GSH level, increase ROS-induced oxidative stress, and decrease the growth of HCT117 colorectal cancer cells in vitro and in vivo, in synergy with SFN and OXP. These findings suggest that QUE in combination with SFN can enhance the susceptibility of colorectal cancer cells to OXP, which would be a promising chemotherapeutic intervention in the treatment of colorectal cancer.

## Figures and Tables

**Figure 1 foods-12-01733-f001:**
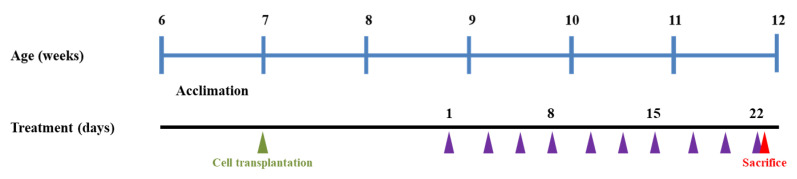
Animal experimental scheme. HCT116 xenograft mice were generated by subcutaneous transplantation (green triangle) in BALB/c nude mice, and treatment doses were intraperitoneally administered three times a week for three weeks (purple triangles, denoting the points at which samples were administered). The size of each xenografted tumor was regularly measured using a caliper until sacrifice (red triangle).

**Figure 2 foods-12-01733-f002:**
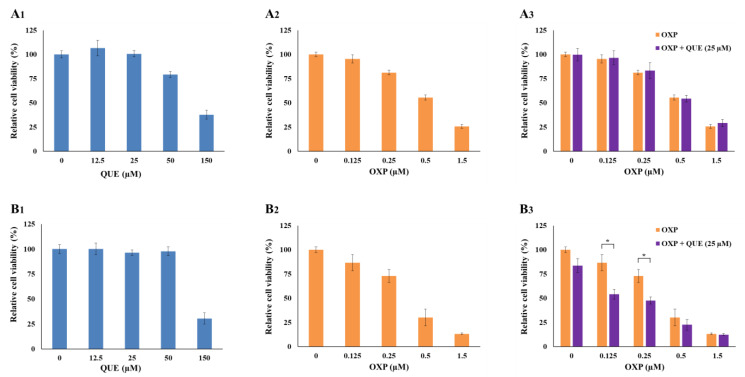
**QUE enhanced OXP-mediated suppression of HCT116 cell proliferation.** HCT116 cells were plated in a 96-well plate and treated with QUE, OXP, or their combination for 3 days (**A_1_–A_3_**) and 6 days (**B_1_–B_3_**). QUE at ≤50 μM for 6 days did not influence cell viability. However, the viability of OXP-treated cells was significantly decreased in the presence of QUE at 25 μM. Error bars, mean ± SEM (*N* = 3). Statistical significance was determined by Student’s unpaired *t*-test at *p* < 0.05, and indicated by an asterisk, *.

**Figure 3 foods-12-01733-f003:**
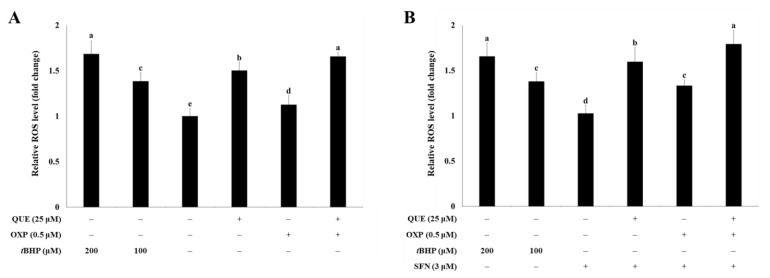
**QUE in combination with OXP synergistically increased the ROS level in HCT116 cells.** HCT116 cells were plated in a black-bottom 96-well plate at a density of 1 × 10^4^ cells per well. After incubating for 24 h, cells were treated with 20 μM H_2_DCFDA for 30 min, followed by incubation with QUE (25 μM), OXP (0.5 μM), both compounds and *t*BHP (100 μM or 200 μM, a positive control for ROS production) in the absence (**A**) or presence (**B**) of SFN for 30 min. Subsequently, the fluorescence was measured. Error bars, mean ± SEM (*N* = 3). Significant differences among the experimental groups are denoted by different alphabetical letters (*p* < 0.05, one-way ANOVA with Duncan’s post hoc HSD test).

**Figure 4 foods-12-01733-f004:**
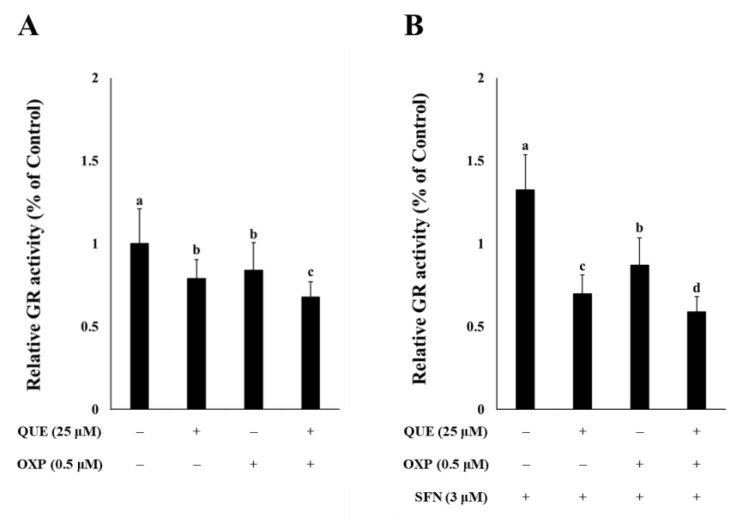
**QUE and SFN in combination with OXP effectively inhibited GR activity in HCT116 cells.** The cells were treated with different combinations of QUE and OXP in the absence (**A**) or presence of SFN (**B**) for 72 h. The GR activity was then measured and normalized to the total protein content. Error bars, mean ± SEM (*N* = 3). Significant differences among treatments were denoted by different alphabetical letters (*p* < 0.05, one-way ANOVA with Duncan’s post hoc HSD test).

**Figure 5 foods-12-01733-f005:**
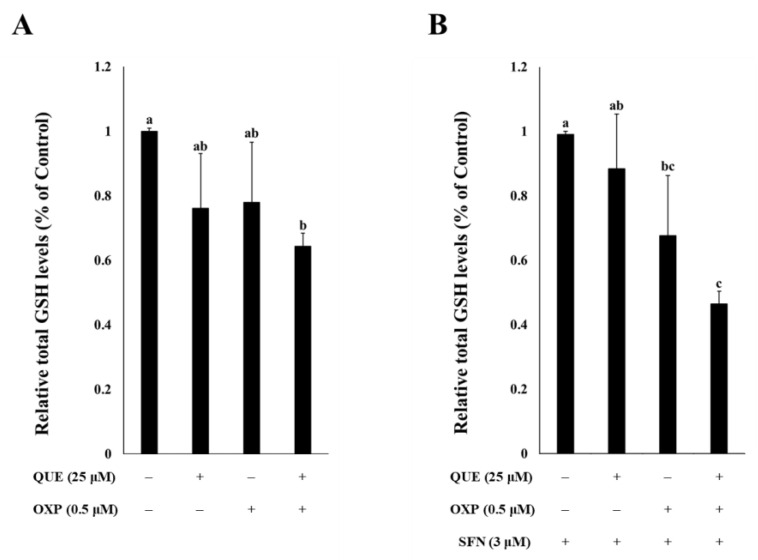
**QUE and SFN in combination with OXP decreased intracellular GSH level in HCT116 cells.** The cells were treated with different combinations QUE and OXP in the absence (**A**) or presence of SFN (**B**) for 72 h. The total GSH levels were then measured and normalized to the total protein content. Error bars, mean ± SEM (*N* = 3). Significant differences among treatments were denoted by different alphabetical letters (*p* < 0.05, one-way ANOVA with Duncan’s post hoc HSD test).

**Figure 6 foods-12-01733-f006:**
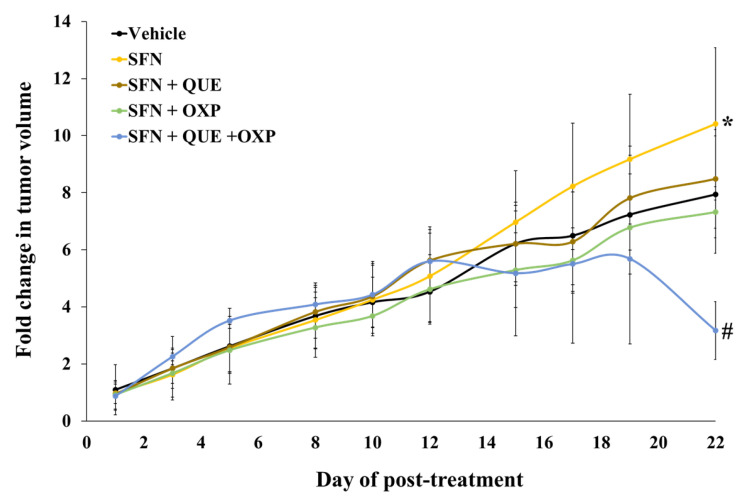
**A combination of QUE, SFN, and OXP suppressed HCT116 xenograft tumor growth.** QUE suppressed the tumor growth of HCT116 cells in vivo. The xenograft tumors were established by subcutaneously transplanting 3 × 10^6^ HCT116 cells to BALB/c mice. The xenograft tumor-bearing mice were treated with QUE (50 mg/kg BW), OXP (10 mg/kg BW), SFN (5 mg/kg BW), and their combinations as designated by intraperitoneal injection. Tumor growth in xenografts was measured three times per week. Error bars represent mean ± SD from four tumor samples (*n* = 4). A significant difference is indicated by asterisk, *, and hashtag, #, compared with vehicle and SFN-treated group (*p* < 0.05, Student’s unpaired *t*-test).

**Figure 7 foods-12-01733-f007:**
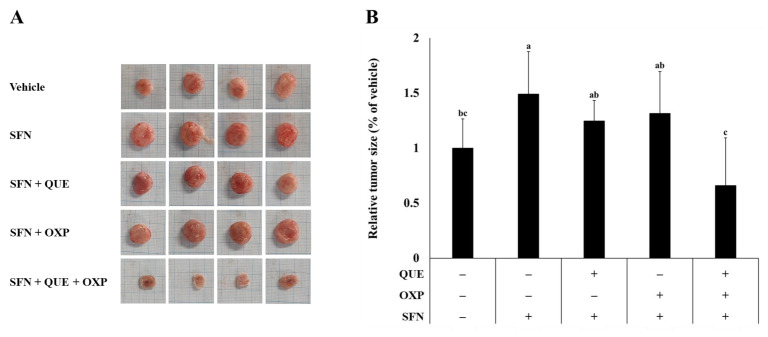
**A combination of QUE, SFN, and OXP synergistically reduced the tumor size in HCT116 xenograft mouse model.** (**A**) The xenograft tumors were dissected on the day of sacrifice. (**B**) The length and width of each tumor were measured using a caliper. Error bars represent mean ± SD (*n* = 4). Different alphabetical letters indicate significant differences among the conditions (*p* < 0.05, one-way ANOVA with Duncan’s post hoc HSD test).

**Figure 8 foods-12-01733-f008:**
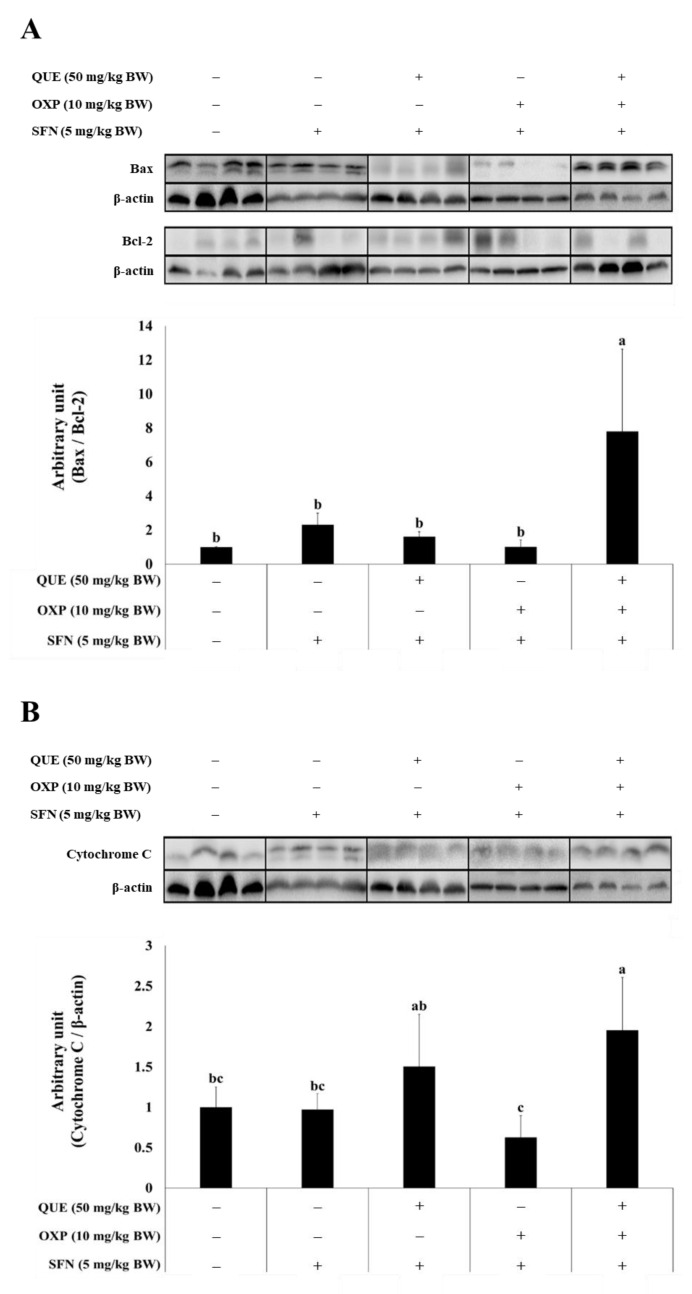
**A combination of QUE, SFN, and OXP increased the protein expressions of apoptotic markers in HCT116 xenograft tumor.** The levels of pro-apoptotic proteins, including Bax/Bcl-2, cytochrome C, cleaved PARP, and cleaved caspase 3, were increased in the xenograft tumors from the mice given QUE in combination with OXP. Western blot images exhibit the expression levels of Bax/Bcl-2 (**A**), cytochrome C (**B**), cleaved PARP/full-length PARP (**C**), and cleaved caspase 3/caspase 3 (**D**). β-actin was used as a loading control to ensure consistency. The quantitative graphs were constructed with data from four independent tumor samples from each treatment group. Error bars represent mean ± SD (*n* = 4). Different alphabetical letters indicate significant differences among the conditions (*p* < 0.05, one-way ANOVA with Duncan’s post hoc HSD test). NS, no statistical significance.

**Figure 9 foods-12-01733-f009:**
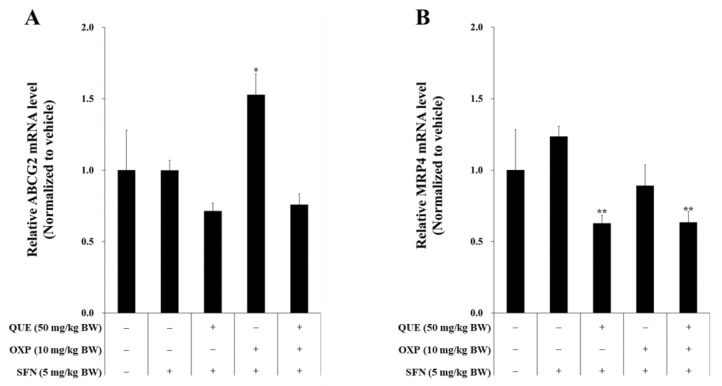
**A combination of QUE, SFN, and OXP downregulated the transcriptional expression of the *ABCG2* and *MRP4* genes in xenograft tumors.** The mRNA expression levels of the drug-resistant genes *ABCG2* (**A**) and *MRP4* (**B**) in HCT116 xenografts were measured using RT-qPCR. Error bars represent means ± SD (*n* = 4). A significant difference is indicated by asterisks compared with vehicle and SFN (*, *p* < 0.05, **, *p* < 0.01, Student’s unpaired *t*-test).

**Table 1 foods-12-01733-t001:** Experimental groups for HCT116 cancer cell xenograft in BALB/c nude mice.

Group	Treatment	
Vehicle	−SFN	Vehicle
SFN	+SFN (5 mg/kg BW)	Vehicle
QUE		QUE at 50 mg/kg BW
OXP		OXP at 10 mg/kg BW
QUE + OXP		QUE at 50 mg/kg BW + OXP at 10 mg/kg BW

−SFN, without SFN administration; +SFN, with SFN administration

## Data Availability

Data is contained within the article.

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
