# Peer review of "Quercetin-Induced Glutathione Depletion Sensitizes Colorectal Cancer Cells to Oxaliplatin"

_foods, 2023, doi:10.3390/foods12081733_

Round 1

Reviewer 1 Report

The study tried to prove that the depletion of reduced glutathione by quercetin and sulforaphane could strengthen the anticancer efficacy of oxaliplatin.

The study is highly innovative, but there are still some deficiencies in conclusion verification.

1.At least two cell lines are needed for tumor research.

2.The paper lacks pictures of animal models.

3. The expression of internal reference protein in most pictures is inconsistent, which greatly affects the interpretation of experimental results.

Author Response

The study tried to prove that the depletion of reduced glutathione by quercetin and sulforaphane could strengthen the anticancer efficacy of oxaliplatin.

The study is highly innovative, but there are still some deficiencies in conclusion verification.

Response: We would like to express our deepest gratitude to the reviewer for taking a precious time and making the constructive comments.

  1. At least two cell lines are needed for tumor research.

Response: This is a follow-up to our previous study by Gwon et al. published in 2020 (refer to the Ref. #15). What we have found and reported was as follows:

  • SFN can activate Nrf2-mmediated antioxidant enzymes in both p53-WT and p53-KO HCT116 colorectal cancer cells, decrease apoptotic protein expression in WT cells but increased in KO cells in a dose-dependent manner, and increase the expression of a mitochondrial biogenesis marker PGC1α in WT cells but decreased in KO cells.
  • A low dose of SFN promoted tumor growth, upregulated the Nrf2 signaling pathway, and decreased apoptotic cell death in p53-WT HCT116 xenografts compared to that in p53-KO HCT116 xenografts in BALB/c nude mice.
  • These observations drove us to conclude that SFN can proliferate HCT116 colorectal cancer cells, which occurs through a crosstalk between the Nrf2 signaling pathway and p53 axis.

Further, the large number of commercial colorectal cell lines available would represent the inter-tumour heterogeneity of the human disease, but in reality, few cell lines lead to reliable primary tumour growth, and fewer still to naturally metastatic CRC. Success rates appear to be highest when using the colorectal carcinoma cell lines HCT-116 or HT-29, but it is not yet clear why this is (McIntyre et al., Bioessays 37: 909–920).

In addition, our unpublished data showed that intracellular GSH level is important for Nrf2-mediated HCT116 cell proliferation. The present study demonstrated that GSH depletion by GR inhibition sensitized HCT116 cells to OXP possibly through oxidative stress-induced apoptotic cell death.

Taken together, it is conceived that the growth of HCT116 cells or their xenograft tumors in mice is regulated through the interplay among the several mediators associated with cellular redox balance and metabolic homeostasis. We are still on the way to solve a riddle and have tried it first in this cell line. Further studies in other cell types are currently ongoing in our lab.

We thank and agree to the referee’s point that the findings should be proven in another cell lines. However, we believe at the same time that the results from this study would be worth publicizing to provide insight into scrutinizing critical intracellular events within the proliferating and/or drug-resistant colorectal cancer cells.

  1. The paper lacks pictures of animal models.

Response: The representative pictures of the dissected tumors were additionally inserted in Figure 7.  

  1. The expression of internal reference protein in most pictures is inconsistent, which greatly affects the interpretation of experimental results.

Response: We understand the referee’s concern that the inconsistent levels of beta-actin among the samples would affect the interpretation of experimental results.

As the referee mentioned, we have experienced the variations among the tissue samples and finally set up a protocol as briefly described in section 2.9. to minimize the variations and to avoid the inconsistency.

In further detail, all the dissected tissues were first homogenated and fractionated at once using the same batch of reagents, solutions, and buffers. Second, the fractionated proteins were quantified by the Bradford assay and the same amount of proteins were loaded onto SDS-PAGE gel. Third, the protein samples of the control group were loaded together with the protein samples from other experimental groups. Fourth, the protein bands were densitometrically analyzed using an image processing software called Image Studio. Fifth, the band of a protein of interest were normalized to its corresponding band of a constitutively expressing protein such as beta-actin. Sixth, the normalized densities of the protein bands from a group were averaged, graphed, and compared with those from other group(s).

Reviewer 2 Report

Comments for authors

The manuscript “Phytochemical-Induced Glutathione Depletion Sensitizes Colorectal Cancer Cells to Oxaliplatin” is very interesting and presented scientifically. The below are comments and suggestions.

1. The title should be specific to “Quercetin” instead of “Phytochemicals”.

2. The authors explain the synergistic effect of quercetin and oxaliplatin, which is initiated by the depletion of glutathione, is the depletion temporary or permanent?

3. Why authors selected Oxaliplatin with quercetin, as there are numerous anticancer drugs?

4. Most of the methods are lacking any literature support. Please provide relevant references to support your protocols.

5. Conclusion should be strengthened.

Author Response

The manuscript “Phytochemical-Induced Glutathione Depletion Sensitizes Colorectal Cancer Cells to Oxaliplatin” is very interesting and presented scientifically. The below are comments and suggestions.

Response: We appreciate the time and effort that the referee has dedicated to providing such valuable feedback on our manuscript.

  1. The title should be specific to “Quercetin” instead of “Phytochemicals”.

Response: We agree with the referee. The title was revised to “Quercetin-Induced Glutathione Depletion Sensitizes Colorectal Cancer Cells to Oxaliplatin”.

  1. The authors explain the synergistic effect of quercetin and oxaliplatin, which is initiated by the depletion of glutathione, is the depletion temporary or permanent?

Response: Thank you for such an interesting comment.

At this point, it is hard to tell if the depletion of GSH by QUE treatment (in combination with SFN) is temporary or permanent, since we have not performed the experiment in which QUE was removed from the culture condition or administration regimen.

Considering the ADME in xenobiotic metabolism, it is unlikely that the phenomenon may be permanent in vivo. However, a further study is required to verify how long the effectiveness will last in a cellular or physiological level.

  1. Why authors selected Oxaliplatin with quercetin, as there are numerous anticancer drugs?

Response:

There are two primary reasons why we have chosen Oxaliplatin (OXP) among numerous anticancer drugs as follows:

  • OXP is a common chemotherapeutic agent for the treatment of colorectal cancer (Graham J., Mushin M., Kirkpatrick P. Oxaliplatin. Rev. Drug Discov. 2004;3:11–12. doi: 10.1038/nrd1287).
  • According to our previous study, OXP was effective to induce cell cycle arrest and further apoptotic cell death in HCT116 cells (Jang, C. H., Moon, N., Oh, J., Kim, J. S. Luteolin Shifts Oxaliplatin-Induced Cell Cycle Arrest at G₀/G₁ to Apoptosis in HCT116 Human Colorectal Carcinoma Cells. Nutrients 2019;11:770. doi: 10.3390/nu11040770).

Metal-based compounds were used since antiquity as treatment for many diseases. However, platinum-based compounds, such as cisplatin, were found to show cytotoxic benefit in cancer unlike any other metal, due to their higher selectivity, lower toxicity, and broader spectrum of activity. In fact, several platinum-analogues were developed since the approval of cisplatin in the late 1970s. Among them, only two compounds, carboplatin and OXP, were globally approved for usage in the clinic. Thus, the three of platinum-based drugs (cisplatin, carboplatin and OXP) have remained as prominent anticancer drugs for the treatment of many solid tumors, including testicular, ovarian, bladder and colorectal cancers.

As briefly addressed in the second paragraph of the Introduction, OXP has been specifically developed for treating colorectal cancer which is resistant and insensitive to cisplatin or carboplatin. This drug is currently a standard in the management of colorectal cancer.

Taken all together, we have decided to use OXP in this study.

  1. Most of the methods are lacking any literature support. Please provide relevant references to support your protocols.

Response: Materials and Methods section was revised accordingly with additional references inserted.

  • 3. section was revised with a couple of references newly inserted.
  • Assays described in 2.4. and 2.5. sections were performed using commercially available assay kits. Thus the catalog number for each kit was provided.
  • 6. section was revised with a couple of references newly inserted.
  • 7. section was revised with a couple of references newly inserted.
  • 8. and 2.9. sections were also revised with additional references inserted.

  1. Conclusion should be strengthened.

Response: Conclusion section was revised accordingly as suggested.

Round 2

Reviewer 1 Report

Thanks very much for your reply. However, I did not see any correction of the irregular reference protein in the pictures.

Author Response

Response to reviewer’s comments

Thanks very much for your reply. However, I did not see any correction of the irregular reference protein in the pictures.

Response: Thank you for your helpful comment. I completely understand your concern about the irregular reference protein in our western blot images. We aimed for consistency in the density of beta actin in the Western blot, but unfortunately, as you pointed out, this is not always achievable due to various factors, such as sample variability and technical limitations.

However, we attempted to load the same amount of protein in each lane and normalized the expression levels of the proteins of interest with the loading control, beta-actin. Therefore, even with irregularities in the beta-actin bands, we can still accurately compare the expression levels of our proteins of interest among the experimental groups.

While we strive for consistency in the Western blot bands, it is worth noting that most journals allow some variability in the intensity of the loading control bands. Nonetheless, we appreciate your feedback and will continue to improve our techniques to achieve greater consistency in future experiments.

For clarity we revised the legend of Figure 8 as follows;

Figure 8. A combination of QUE, SFN and OXP increased the protein expressions of apoptotic markers in HCT116 xenograft tumor. The levels of pro-apoptotic proteins, Bax/Bcl-2 (A), cyto-chrome C (B), cleaved PARP (C), and cleaved caspase 3 (D), were assessed using western blot analysis on four independent samples from each treatment group. To ensure consistency, β-actin was used as a loading control. The quantitative graphs show the mean ± standard deviation (SD) of the WB image density ratios of the proteins of interest relative to β-actin, the loading control, from four tumor samples per group. Different alphabetical letters indicate significant differences among the conditions (p < 0.05, one-way ANOVA with Duncan’s post-hoc HSD test).

Here are some references;

  1. Du et al., Cell Death and Disease (2022) 13:187
  2. Desnoyers et al., Oncogene (2008) 27, 85–97
  3. Voloshanenko et al., NATURE COMMUNICATIONS 4:2610 (2013)